# SLC1A5 provides glutamine and asparagine necessary for bone development in mice

**Deepika Sharma[1], Yilin Yu[2], Leyao Shen[2], Guo-Fang Zhang[3,4], Courtney M Karner[1,2,5]***

[1]Department of Orthopaedic Surgery, Duke University School of Medicine, Durham, United States; [2]Department of Internal Medicine, University of Texas Southwestern Medical Center, Dallas, United States; [3]Sarah W. Stedman Nutrition and Metabolism Center & Duke Molecular Physiology Institute, Duke University Medical Center, Durham, United States; [4]Department of Medicine, Duke University School of Medicine, Durham, United States; [5]Charles and Jane Pak Center for Mineral Metabolism and Clinical Research. University of Texas Southwestern Medical Center at Dallas, Dallas, United States

**Abstract** Osteoblast differentiation is sequentially characterized by high rates of proliferation followed by increased protein and matrix synthesis, processes that require substantial amino acid acquisition and production. How osteoblasts obtain or maintain intracellular amino acid production is poorly understood. Here, we identify SLC1A5 as a critical amino acid transporter during bone development. Using a genetic and metabolomic approach, we show SLC1A5 acts cell autonomously to regulate protein synthesis and osteoblast differentiation. SLC1A5 provides both glutamine and asparagine which are essential for osteoblast differentiation. Mechanistically, glutamine and to a lesser extent asparagine support amino acid biosynthesis. Thus, osteoblasts depend on *Slc1a5* to provide glutamine and asparagine, which are subsequently used to produce non-essential amino acids and support osteoblast differentiation and bone development.

**\*For correspondence:**
courtney.karner@
utsouthwestern.edu

**Competing interest:** The authors declare that no competing interests exist.

## Introduction

Osteoblasts are secretory cells responsible for producing and secreting the Collagen Type one rich extracellular bone matrix. Osteoblasts differentiate from mesenchymal progenitors in a well-coordinated temporal sequence regulated by the transcription factors RUNX2 and OSX (encoded by *Sp7*) (*Nakashima et al., 2002*; *Otto et al., 1997*). Osteoblast progenitors are proliferative before undergoing terminal differentiation into postmitotic Collagen Type 1 (COL1A1) matrix producing osteoblasts (*Quarles et al., 1992*; *Pardee et al., 1978*; *Tonna and Cronkite, 1961*; *Stein et al., 1990*; *Stein and Lian, 1993*). Both proliferation and matrix production are biosynthetically demanding and burden osteoblasts with enhanced metabolic demands. For example, proliferation requires cells to increase nutrient and amino acid acquisition to generate the biomass necessary to duplicate cell mass for division. Likewise, matrix production places similar demands upon osteoblasts. Thus, bone formation is associated with increased synthetic demands due to osteoblast proliferation, differentiation, and bone matrix production (*Guntur et al., 2014*; *Lee et al., 2017*; *Karner and Long, 2018*; *Riddle and Clemens, 2017*).

These enhanced synthetic demands are predicted to require a constant supply of amino acids to sustain both proliferation and bone matrix production. Cells can obtain amino acids via uptake from the extracellular milieu or from de novo synthesis from glucose or other amino acids (*Karner and*

*Long, 2018*; *Flanagan and Nichols, 1964*; *Adamson and Ingbar, 1967*; *Finerman and Rosenberg, 1966*; *Yee, 1988*; *Palm and Thompson, 2017*). In osteoblasts, amino acid consumption is known to be transcriptionally regulated (*Elefteriou et al., 2006*; *Rached et al., 2010*) and rapidly increases during differentiation and in response to osteoinductive signals (*Karner et al., 2015*; *Shen et al., 2021*; *Yu et al., 2019*; *Hahn et al., 1969*; *Phang and Downing, 1973*). Importantly, genetic mutations that limit amino acid uptake are associated with decreased proliferative capacity and reduced bone formation (*Elefteriou et al., 2006*; *Rached et al., 2010*; *Hu et al., 2020*). Despite this, little is known about how osteoblasts obtain the amino acids necessary to promote robust proliferation and bone matrix synthesis.

The regulation of amino acid supply is an important regulatory node that is frequently upregulated to support proliferation and biosynthesis in many pathological conditions (*Garcia-Bermudez et al., 2018*; *Jones et al., 2019*; *Bhutia et al., 2015*; *Bröer et al., 2016*). Amino acid uptake is controlled by a diverse array of membrane-bound transport proteins that transport amino acids into and out of the cell. Once inside the cell, amino acids have diverse fates. For example, amino acids may contribute directly to protein synthesis, facilitate signaling or be metabolized to generate ATP or other inter-mediate metabolites including other amino acids (*DeBerardinis et al., 2007*; *Green et al., 2016*; *Wang et al., 2009*; *Karner and Long, 2017*). How osteoblasts obtain or utilize amino acids is not well understood. We recently identified Alanine, Serine, Cysteine transporter 2 (ASCT2, denoted herein as SLC1A5, encoded by *Slc1a5*) as a potential regulator of amino acid supply in osteoblast progenitors (*Shen et al., 2021*; *Hu et al., 2020*). SLC1A5 is a $Na^+$-dependent neutral amino acid exchanger that can transport glutamine, alanine, serine, asparagine, and threonine (*Bröer et al., 1999*; *Scopelliti et al., 2018*; *Utsunomiya-Tate et al., 1996*; *Nakaya et al., 2014*; *Wu et al., 2021*). In cancer cells, *Slc1a5* upregulation is associated with metabolic reprograming and is necessary for increased prolif-eration and biosynthesis (*Bröer et al., 2016*; *Liu et al., 2018*; *Ren et al., 2015*; *van Geldermalsen et al., 2016*; *Wang et al., 2014*; *Schulte et al., 2018*; *Hassanein et al., 2013*). In comparison, little is known about the role of SLC1A5-dependent amino acid uptake during bone development.

Here, we define the role of amino acid uptake through *Slc1a5* during bone development and homeostasis. Using a genetic and metabolic approach, we demonstrate *Slc1a5* is required for osteo-blast proliferation, differentiation and bone matrix production. Mechanistically, SLC1A5 provides glutamine and asparagine to maintain intracellular amino acid homeostasis in osteoblasts. Collec-tively, these data highlight the previously unknown role for *Slc1a5* in osteoblasts regulating differen-tiation and bone development.

## Results

### *Slc1a5* is required for bone development

We previously identified *Slc1a5* as a potential regulator of proliferation and WNT induced osteo-blast differentiation in stromal cells (*Shen et al., 2021*; *Hu et al., 2020*). To understand if *Slc1a5* functions in osteoblasts, we first characterized *Slc1a5* expression during osteoblast differentiation. *Slc1a5* is highly expressed in both naive calvarial cells and bone marrow stromal cells and signifi-cantly increases during osteoblast differentiation (*Figure 1A*). To determine if *Slc1a5* is required for osteoblast differentiation, we first utilized a CRISPR/Cas9 approach to knock out *Slc1a5* in cultured calvarial cells (*Figure 1—figure supplement 1A*). Western blot analyses confirmed this approach effectively ablated SLC1A5 protein (*Figure 1—figure supplement 1B*). *Slc1a5* targeting did not affect early osteoblast differentiation but did prevent the induction of the mature osteoblast genes *Ibsp* and *Bglap* and prevented matrix mineralization in primary calvarial cells (*Figure 1B–C*). To determine if *Slc1a5*-dependent amino acid uptake is important for osteoblast differentiation in vivo, we generated a conditional (floxed) *Slc1a5* allele (*Slc1a5*[fl]) using homologous recombina-tion (*Figure 1—figure supplement 1C*). These mice were crossed with the *Sp7tTA;tetOeGFP/Cre* deleter mouse (denoted hereafter as *Sp7Cre*) that expresses GFP and Cre recombinase under the control of the *Sp7* promoter to generate mice lacking *Slc1a5* in committed osteoblast progenitors (*Rodda and McMahon, 2006*). Western blot analysis confirmed the specific ablation of SLC1A5 in bones isolated from *Sp7Cre;Slc1a5*[fl/fl] mice (*Figure 1—figure supplement 1D*). *Sp7Cre;Slc1a5*[fl/fl] mice were characterized by delayed endochondral and intramembranous ossification evident at both embryonic (E) day E14.5 and E15.5 (*Figure 1D–E–*). In addition to delayed mineralization,

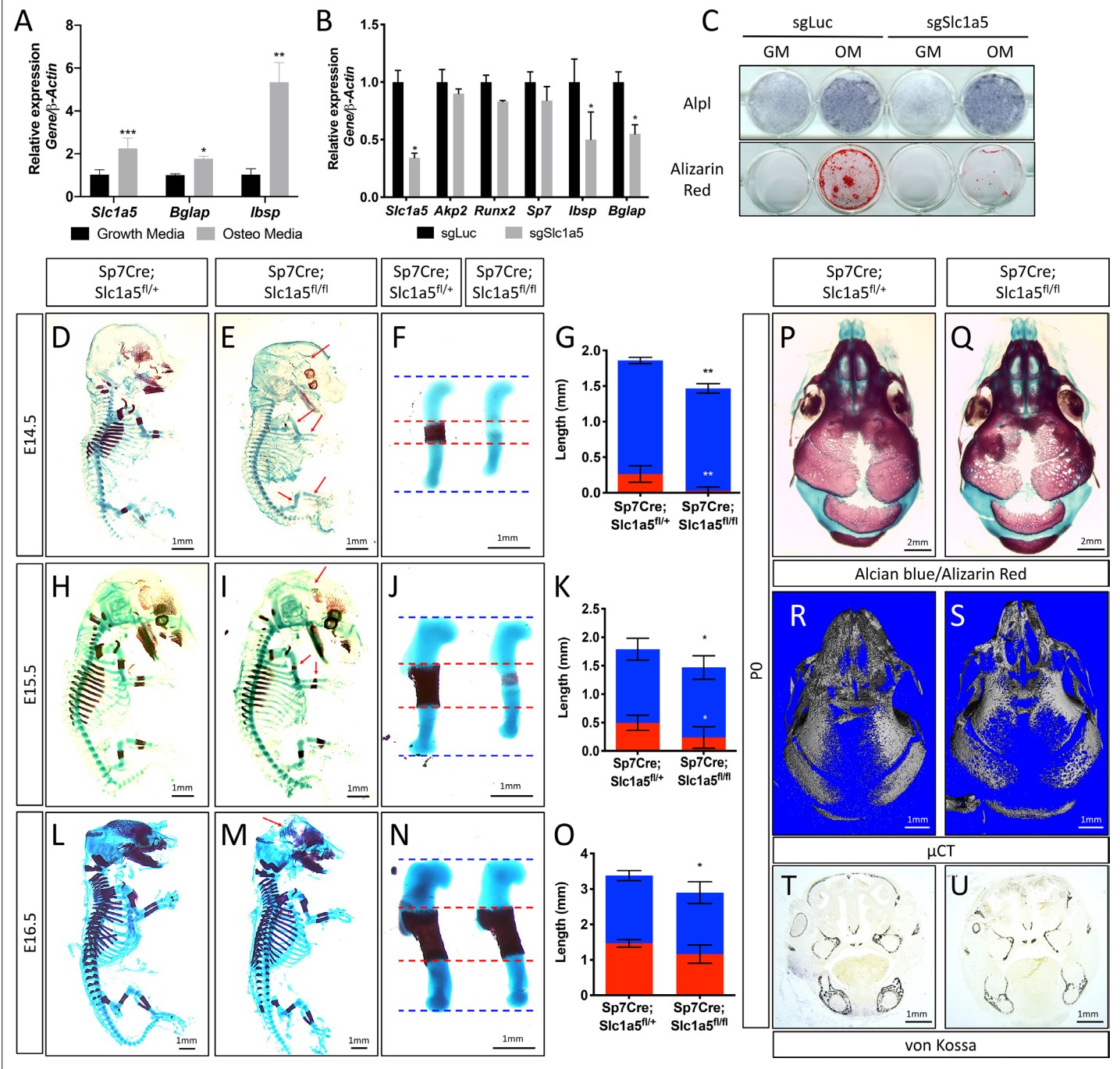

**Figure 1.** Slc1a5 is required for bone development in mice. (**A**) qRT-PCR analyses of gene expression in calvarial osteoblasts (cOB) cultured for 7 days in growth or osteogenic media. (**B–C**) qRT-PCR analyses (**B**) or functional assays (**C**) of the effect of *Slc1a5* deletion on osteoblast differentiation in cOB cultured for 7 days in osteogenic media. (**D–O**) Skeletal preparations of *Sp7Cre;Slc1a5*^fl/+ and *Sp7Cre;Slc1a5*^fl/fl mice at E14.5 (n = 5), E15.5 (N = 8) and E16.5 (N = 5). Arrows denote reduced mineralization. Isolated humeri shown in (**F**), (**J**) and (**N**). Blue dotted lines denotes the control overall humerus length. Red dotted lines denote control mineralized area. Images quantified in (**G, K and O**). (**P–S**) A representative skeletal preparation (**P–Q**) or (**R–S**) Representative Micro-computed tomography (μCT) (N = 6) used to quantify BV/TV(%) and (**T–U**) von Kossa staining on *Sp7Cre;Slc1a5*^fl/+ and *Sp7Cre;Slc1a5*^fl/fl (N = 4) knockout mice at P0. Error bars depict SD. * p ≤ 0.05, ** p ≤ 0.005, *** p ≤ 0.0005, **** p ≤ 0.00005, by an unpaired two-tailed Student's *t*-test.

The online version of this article includes the following figure supplement(s) for figure 1:

**Source data 1.** Contains numerical source data for *Figure 1*.

**Figure supplement 1.** Slc1a5 ablation decreases bone development.

*Figure 1 continued on next page*

*Figure 1 continued*

**Figure supplement 1—source data 1.** Contains numerical and uncropped western blot source data for *Figure 1—figure supplement 1*.

**Figure supplement 2.** Skull phenotype at 2 months of age in *Sp7Cre;Slc1a5fl/fl* mice.

*Sp7Cre;Slc1a5^{fl/fl}* mice had reduced size of various skeletal elements (exemplified by the humerus, *Figure 1F and J*) relative to *Sp7Cre;Slc1a5^{fl/+}* littermate controls (*Figure 1F–G and J–K* and *Figure 1—figure supplement 1E-L*). By E16.5 the extent of mineralization was no longer significantly different although the overall length of individual skeletal elements was significantly different from *Sp7Cre;Slc1a5^{fl/+}* littermate controls (*Figure 1L–O* and *Figure 1—figure supplement 1M-P*). At birth, neither matrix mineralization or nor element length were significantly different suggesting there is a transient delay in endochondral ossification (*Figure 1—figure supplement 1Q-X*). Conversely, *Sp7Cre;Slc1a5^{fl/fl}* mice are characterized by impaired intramembranous ossification at all stages evaluated (*Figure 1D–E–, L–M and P–Q* and *Figure 1—figure supplement 1Q-R*). At birth, *Sp7Cre;Slc1a5^{fl/fl}* calvariae had increased porosity (1.0 ± 0.2 vs 2.3 ± 0.2 in *Sp7Cre;Slc1a5^{fl/+}* and *Sp7Cre;Slc1a5^{fl/fl}* respectively, $p < 0.05$), increased suture width (1.0 ± 0.2 mm vs 1.4 ± 0.3 in *Sp7Cre;Slc1a5^{fl/+}* and *Sp7Cre;Slc1a5^{fl/fl}* respectively, $p < 0.05$) and significantly less bone volume (0.7% ± 0.02% vs 0.6 ± 0.05% in *Sp7Cre;Slc1a5^{fl/+}* and *Sp7Cre;Slc1a5^{fl/fl}* respectively, $p < 0.05$) as measured by micro-computed tomography (µCT) (*Figure 1*). At 2-months of age, *Sp7Cre;Slc1a5^{fl/fl}* mice presented with less bone mass, increased suture width and altered suture morphology (*Figure 1—figure supplement 2*). Von Kossa staining of histological sections confirmed the reduction in mineralized bone matrix in newborn *Sp7Cre;Slc1a5^{fl/fl}* calvariae (*Figure 1T–U*). Thus, *Slc1a5* expression in osteoprogenitors is essential for bone development.

### *Slc1a5* is required for osteoblast differentiation and proliferation

To determine how *Slc1a5* regulates bone development, we first characterized the cellular effects of *Slc1a5* ablation on osteoblasts. Because we observed a consistent skull phenotype at all stages and *Sp7Cre* is expressed in both osteoblasts and hypertrophic chondrocytes in the developing limb (*Rodda and McMahon, 2006*), we focused our analyses on the skull which is formed by intramembranous ossification and does not involve a cartilaginous intermediate (*Ornitz and Marie, 2002*). Von Kossa staining confirmed there was delayed mineralization in *Sp7Cre;Slc1a5^{fl/fl}* calvariae at E15.5 (*Figure 2A–D*). This was not due to changes in the number of osteoblast progenitors as we observed no significant difference in the number of *Sp7^{GFP}* expressing osteoblast progenitors per bone area in *Sp7Cre;Slc1a5^{fl/fl}* mice compared to *Sp7Cre;Slc1a5^{fl/+}* wild type littermates (75.3% ± 11.6% vs 64.7±9.5 % in *Sp7Cre;Slc1a5^{fl/+}* and *Sp7Cre;Slc1a5^{fl/f}* animals, respectively) (*Figure 2E–F*). However, we did observe a significant reduction in the proportion of Sp7^{GFP} cells that were positive for proliferating cell nuclear antigen (PCNA) (25.9% ± 4.6% vs 15.6±2.9 % in *Sp7Cre;Slc1a5^{fl/+}* or *Sp7Cre;Slc1a5^{fl/fl}* respectively, $p < 0.05$) (*Figure 2G–J*) suggesting *Slc1a5* is required for proliferation of *Sp7* expressing osteoblast progenitors. We next evaluated osteoblast differentiation using in situ hybridization. We did not observe significant differences in the expression of early osteoblast genes including alkaline phosphatase (Alpl, as determined by in situ staining) or *Col1a1* (*Figure 2K–N*). Conversely, *Sp7Cre;Slc1a5^{fl/fl}* animals had reduced expression of the osteoblast differentiation genes *Spp1* and *Ibsp* at both E15.5 and postnatal (P) day 0 (P0) (*Figure 2O–T* and *Figure 2—figure supplement 1A-B*). Similarly, the mature osteoblast gene *Bglap* was highly reduced at P0 in *Sp7Cre;Slc1a5^{fl/fl}* compared to *Sp7Cre;Slc1a5^{fl/+}* littermates (*Figure 2U–V*). Consistent with these observations, primary calvarial cells isolated from *Sp7Cre;Slc1a5^{fl/fl}* mice incorporated less EdU and were characterized by reduced osteoblast differentiation and matrix mineralization in vitro (*Figure 2W–X*). This is likely a direct effect of loss of *Slc1a5* activity as acute SLC1A5 inhibition using GPNA reduced calvarial cell proliferation in vitro (*Figure 2—figure supplement 1W*). It is important to note we observed similar results in the developing long bones at both e15.5 and P0 (*Figure 2—figure supplement 1C-V* and *Figure 2—figure supplement 2A-I*). Similarly, 2 -month-old *Sp7Cre;Slc1a5^{fl/fl}* mice had reduced osteoblast numbers (exemplified by OCN expression) and significantly less trabecular bone volume as measured by µCT in the distal femur compared to *Sp7Cre;Slc1a5^{fl/+}* wild -type littermate controls (*Figure 2—figure supplement*

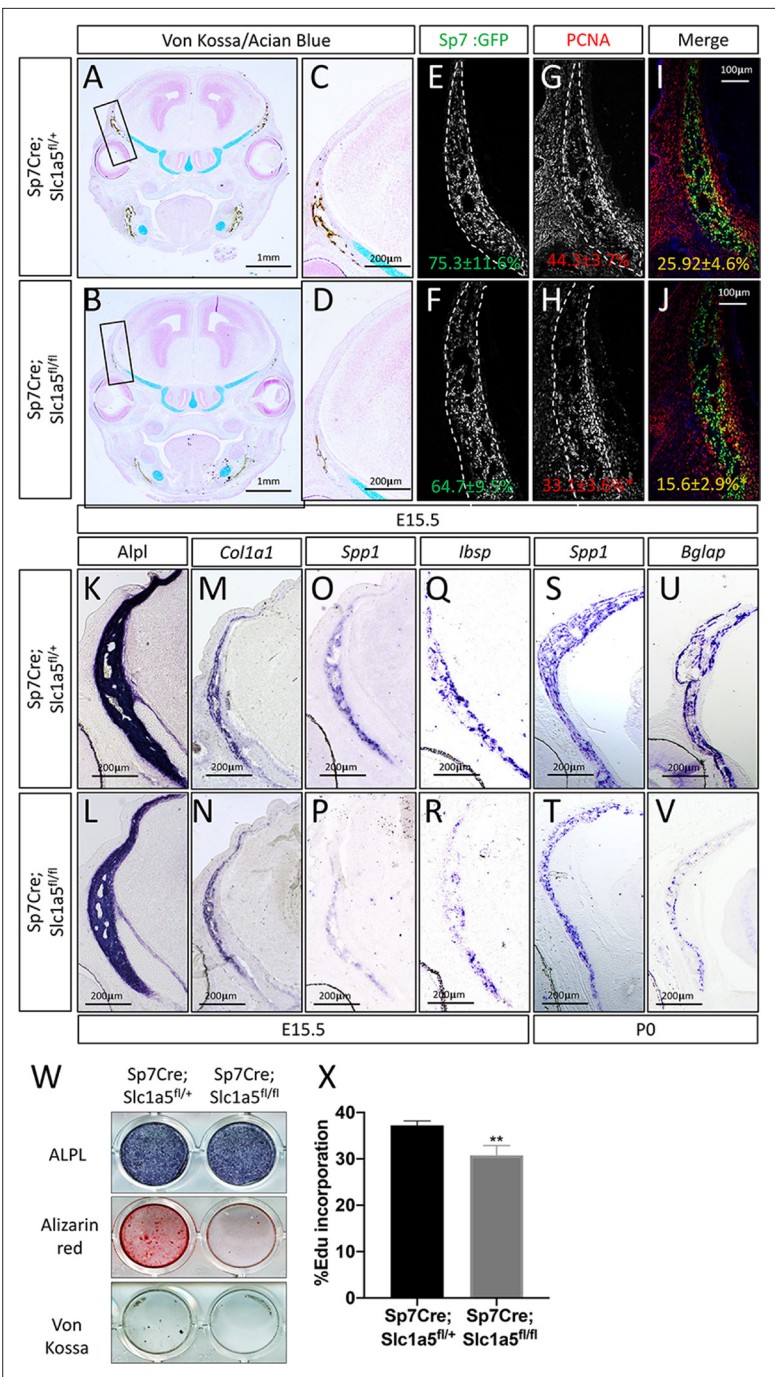

**Figure 2.** Slc1a5 is necessary for osteoblast proliferation and differentiation. (**A–D**) von Kossa/Alcian blue staining on *Sp7Cre;Slc1a5^{fl/+}* (**A,C**) and *Sp7Cre;Slc1a5^{fl/fl}* (**B,D**) (N = 4) at E15.5. (**E–J**) Representative immunofluorescent staining for Proliferating Cell Nuclear Antigen (PCNA) used to quantify proliferation. Endogenous GFP from *Sp7^{GFP}* shown in (**G–H**) used to quantify PCNA/GFP double positive cells. The numbers in each panel represent the percent GFP, PCNA or double positive cells per GFP positive bone area (dotted line). (**K–V**) Representative alkaline phosphatase (ALPL) staining (**K–L**) or In situ hybridization (**M–V**) for *Col1a1, Spp1, Ibsp* at E15.5 (N = 4) and *Spp1, Bglap* at P0 (N = 3). (**W**) Functional assays of osteoblast differentiation in cOB isolated from *Sp7Cre;Slc1a5^{fl/+}* and *Sp7Cre;Slc1a5^{fl/fl}* mice. cultured for 14 days in osteogenic media. (**X**) Graphical depiction of EdU incorporation in cOB cells isolated from *Sp7Cre;Slc1a5^{fl/+}* and *Sp7Cre;Slc1a5^{fl/fl}* mice. Error bars depict SD. * p ≤ 0.05, ** p ≤ 0.005. by an unpaired two-tailed Student's *t*-test.

The online version of this article includes the following figure supplement(s) for figure 2:

*Figure 2 continued on next page*

*Figure 2 continued*

**Source data 1.** Contains numerical source data for *Figure 2*.

**Figure supplement 1.** SLC1A5 is necessary for proliferation and endochondral ossification.

**Figure supplement 1—source data 1.** Contains numerical source data for *Figure 2—figure supplement 1*.

**Figure supplement 2.** Limb phenotypes at birth and 2 months in *Sp7Cre;Slc1a5fl/fl* mice.

**Figure supplement 2—source data 1.** Contains numerical source data for *Figure 2—figure supplement 2*.

*2J-N*). Collectively, these data indicate *Slc1a5* acts cell-autonomously to regulate osteoprogenitor proliferation and osteoblast differentiation.

## *Slc1a5* inhibition reduces protein synthesis in osteoblasts

We next sought to understand how *Slc1a5* regulates proliferation and differentiation. Because *Slc1a5* encodes a neutral amino acid transporter, we hypothesized that *Slc1a5* ablation would primarily affect protein synthesis. To test this hypothesis, we first evaluated protein synthesis directly by monitoring the incorporation of $^3$H-proline into protein. *Slc1a5* targeting significantly reduced $^3$H-proline incorporation into both collagen and total protein (*Figure 3—figure supplement 1A-B*). Likewise, primary calvarial cells isolated from *Sp7Cre;Slc1a5fl/fl* mice had significantly reduced protein and collagen synthesis rates in vitro (*Figure 3A–B*). These results indicate *Slc1a5* is required for robust protein synthesis in osteoblasts. To test the validity of this conclusion, we evaluated the production of osteoblast proteins in vivo. Consistent with the in vitro data, *Sp7Cre;Slc1a5fl/fl* mice were characterized by significantly less COL1A1 in both the calvariae and long bones at e15.5 and P0 (*Figure 3C–H* and *Figure 2—figure supplement 1Q-R* and *Figure 3—figure supplement 1E-F*). It is important to note, *Col1a1* mRNA expression was not affected in these mice indicating *Slc1a5* provides amino acids required for robust COL1A1 synthesis. Similarly, *Sp7Cre;Slc1a5fl/fl* mice had a significant reduction in OSX protein expression as we observed a reduction in the proportion of OSX expressing cells despite no change in *Sp7GFP* expression (used as a proxy for *Sp7*) in *Sp7Cre;Slc1a5fl/fl* calvariae (*Figure 3I–N*). Thus, *Slc1a5* is essential for the synthesis of proteins like OSX to regulate terminal osteoblast differentiation and COL1A1 necessary for bone matrix production.

## *Slc1a5* provides glutamine and asparagine to regulate amino acid homeostasis

We next sought to define the molecular substrates of SLC1A5 in osteoblasts. First, we determined the effect of *Slc1a5* knockout on downstream metabolites using mass spectrometry. *Slc1a5* targeting significantly diminished the intracellular abundance of many amino acids including reported substrates of SLC1A5 (e.g. asparagine, glutamine, and alanine) as well as amino acids not known to be transported by SLC1A5 (e.g. glutamate, lysine, histidine, aspartate, glycine, and proline) (*Figure 4A*). Moreover, *Slc1a5* deletion also reduced the abundance of select TCA cycle intermediates including fumarate, malate, citrate, and a-ketoglutarate (*Figure 4—figure supplement 1A*). Interestingly, the uptake of many of these amino acids was unaffected in *Slc1a5*-deficient cells as only glutamine and to a lesser extent asparagine uptake was diminished in *Slc1a5* targeted calvarial cells (*Figure 4B*). Conversely, we observed a compensatory increase in the uptake of lysine in *Slc1a5*-deficient calvarial cells (*Figure 4B*). Similar results were obtained upon acute SLC1A5 inhibition indicating SLC1A5 transports glutamine and asparagine in osteoblasts (*Figure 4—figure supplement 1B*). *Slc1a5*-deficient cells had many cellular changes consistent with decreased amino acid concentrations. For example, *Slc1a5* inhibition significantly increased EIF2a Ser51 phosphorylation, a marker of amino acid depletion (*Figure 4C* and *Figure 4—figure supplement 1C*). Additionally, we observed a significant reduction in mTORC1 activity, as both ribosomal protein S6 Ser240/244 phosphorylation and Eif4ebp1 Ser65 phosphorylation were significantly reduced in *Slc1a5*-deficient calvarial cells (*Figure 4C*). Interestingly, mTOR activation was not affected by acute SLC1A5 inhibition indicating decreased mTORC1 signaling may be a secondary effect of *Slc1a5* deletion. Collectively, these data indicate SLC1A5 provides glutamine and asparagine to regulate intracellular amino acid homeostasis in osteoblast progenitors.

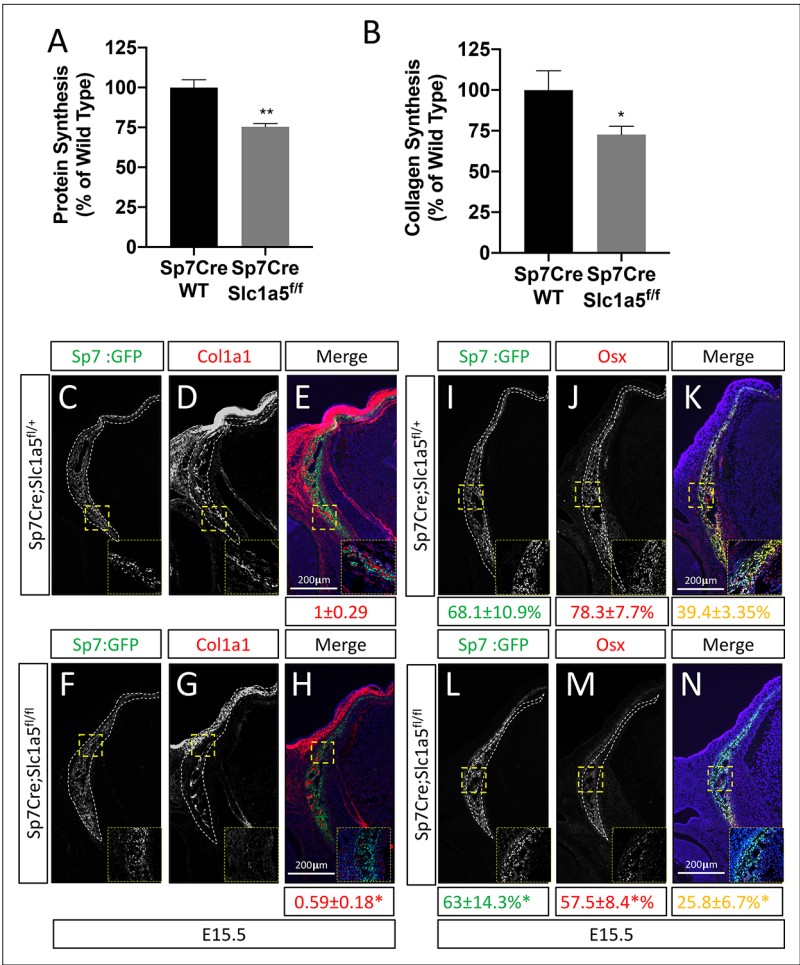

**Figure 3.** Slc1a5 is required for robust protein and matrix synthesis in osteoblasts. (**A–B**) Radiolabeled $^3$H-proline incorporation into total protein (**A**) or collagen (**B**) in cOB cells isolated from *Sp7Cre;Slc1a5*$^{fl/+}$ and *Sp7Cre;Slc1a5*$^{fl/fl}$ mice. (**C–N**) Representative immunofluorescent staining for Collagen Type 1 (COL1A1) (**C–H**) or OSX (**I–N**) at E15.5 in *Sp7Cre;Slc1a5*$^{fl/+}$(**C,D,E,I,J,K**) and *Sp7Cre;Slc1a5*$^{fl/fl}$ (**F,G,H,L,M,N**) mice. Endogenous GFP from *Sp7*$^{GFP}$ shown in (**C,F,I,L**). Col1a1 intensity was quantified in the GFP-positive region. Endogenous GFP from *Sp7*$^{GFP}$ shown in (**I, L**) was used to quantify OSX/GFP double positive cells. The numbers in each panel represent the percent GFP, OSX or double positive cells per GFP-positive bone area (dotted line). Inset images show 60 x magnification of the indicated region. * p ≤ 0.05, by an unpaired two-tailed Student's *t*-test.

The online version of this article includes the following figure supplement(s) for figure 3:

**Source data 1.** Contains numerical source data for *Figure 3*.

**Figure supplement 1.** SLC1A5 is necessary for protein synthesis.

**Figure supplement 1—source data 1.** Contains numerical source data for *Figure 3—figure supplement 1*.

## Glutamine depletion mimics the effects of *Slc1a5* deletion

We next sought to understand the importance of both glutamine and asparagine for cellular function. To do this, we cultured naive calvarial cells in the absence of glutamine or in media treated with asparaginase to deplete asparagine. Depletion of either glutamine or asparagine from the media similarly inhibited the induction of terminal osteoblast genes *Ibsp* and *Bglap* and prevented matrix mineralization (*Figures 1B–C , and 5A–B*). While this was reminiscent of *Slc1a5* ablation, it is important to note that glutamine withdrawal had a more profound effect on osteoblast differentiation compared to either asparagine depletion or *Slc1a5* ablation. We next evaluated mTORC1 activity and EIF2a phosphorylation. Depletion of glutamine, but not asparagine, mimicked the effects of *Slc1a5* targeting on both EIF2a Ser51 phosphorylation and mTORc1 activity (*Figure 5C*). Consistent with this, cells cultured

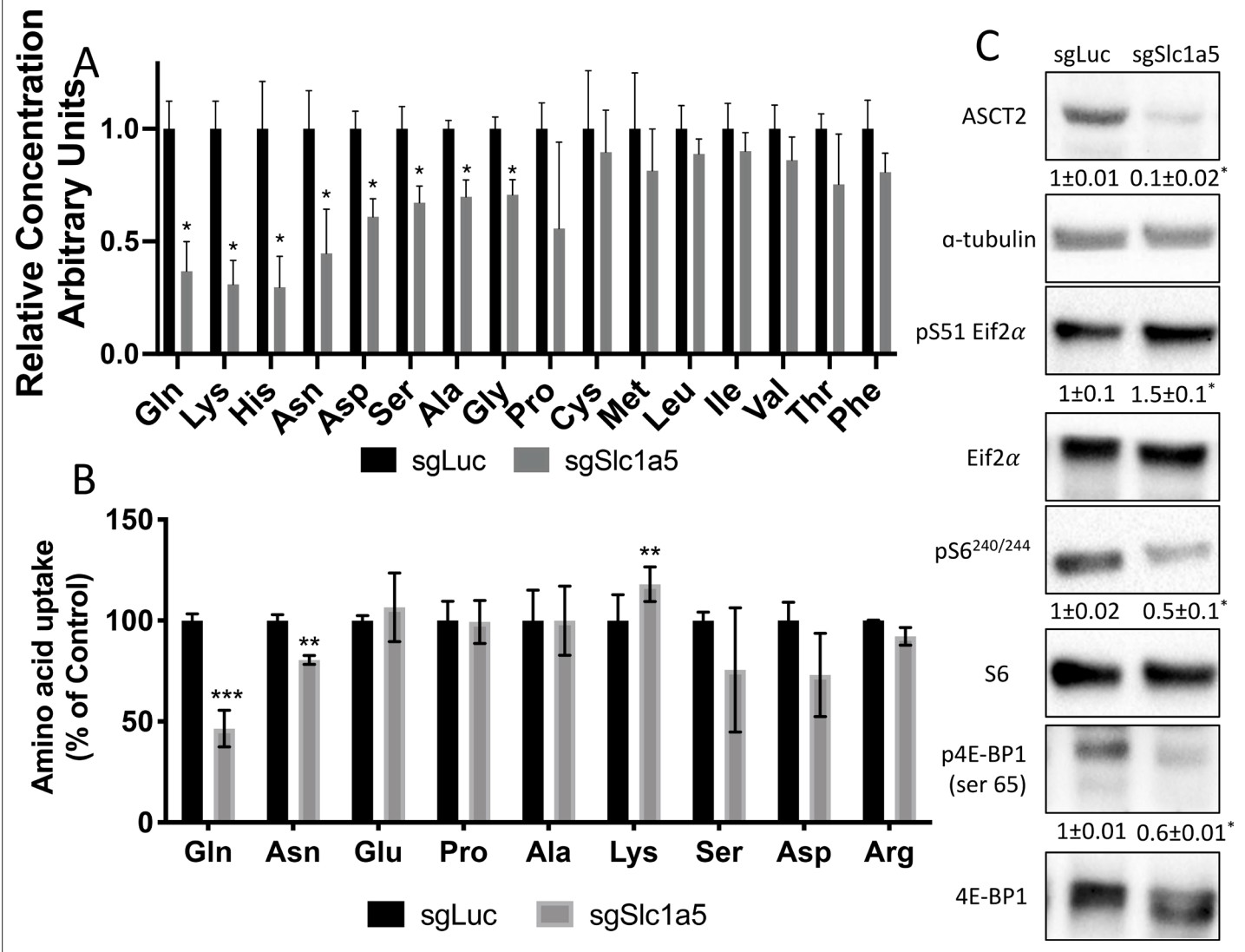

**Figure 4.** Slc1a5 provides glutamine and asparagine to maintain amino acid homeostasis. (**A**) Effect of *Slc1a5* targeting on intracellular amino acid concentration measured by mass spectrometry.(**B**) Effect of *Slc1a5* targeting on the uptake of indicated radiolabeled amino acids. (**C**) Western Blot analyses of the effect of *Slc1a5* targeting on mTORC1 signaling and Eif2a phosphorylation. Phospho-proteins normalized to respective total protein. ASCT2 normalized to α-tubulin. sgRNAs targeting luciferase were used as a negative control. Fold change± SD for *sgSlc1a5* over *sgLuc* in three independent experiments. * p ≤ 0.05 by an unpaired two-tailed Student's *t*-test.

The online version of this article includes the following figure supplement(s) for figure 4:

**Source data 1.** Contains numerical and uncropped western blot source data for *Figure 4*.

**Figure supplement 1.** NEAA can partially rescue differentiation but not proliferation.

**Figure supplement 1—source data 1.** Contains numerical and uncropped western blot source data for *Figure 4—figure supplement 1*.

in the absence of glutamine had reduced COL1A1 expression and decreased proliferation like *Slc1a5* deficient cells (***Figure 5C–D***). On the other hand, culturing cells in the absence of asparagine had no discernable effect on either pSer51 Eif2a, pSer240/244 S6rp, or COL1A1 expression and enhanced proliferation as determined by increased EdU incorporation (***Figure 5C–D***). Importantly, culturing cells in the absence of either glutamine or asparagine had no effect on cell viability (***Figure 5E***). From these data, we conclude SLC1A5 primarily provides glutamine to regulate amino acid homeostasis necessary for proliferation and osteoblast differentiation. Additionally, SLC1A5 provides asparagine which is essential for terminal osteoblast differentiation and matrix mineralization.

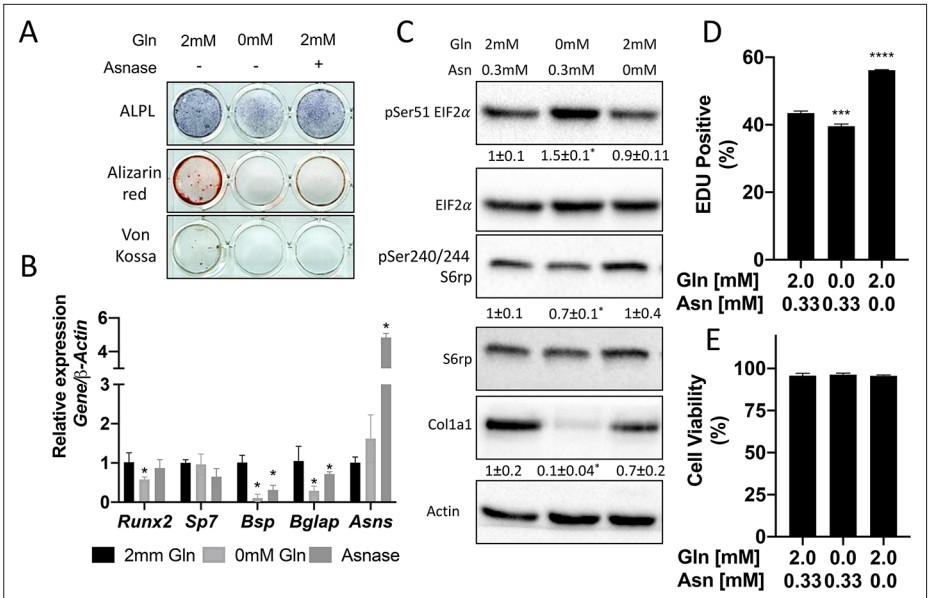

**Figure 5.** Glutamine and asparagine are required for osteoblast differentiation. (**A–B**) Functional assays (**A**) or qRT-PCR analyses (**B**) of the effect of glutamine withdrawal or asparaginase treatment on cOB cultured in osteogenic media for 14 days. (**C**) Western blot analyses of the effect of glutamine or asparagine withdrawal on mTORC1 signaling, Eif2a phosphorylation or COL1A1 expression. Phospho-proteins normalized to respective total protein. COL1A1 normalized to beta-actin. Fold change± SD for three independent experiments. * p ≤ 0.05 by an unpaired 2-tailed Student's *t*-test. (**D–E**) Effect of glutamine or asparagine withdrawal on EdU incorporation (**D**) or cell viability (**E**) as determined by Annexin V staining.

The online version of this article includes the following figure supplement(s) for figure 5:

**Source data 1.** Contains numerical and uncropped western blot source data for *Figure 5*.

## Glutamine and asparagine-dependent amino acid synthesis is essential for protein synthesis

We next investigated how osteoblast utilize glutamine and asparagine. Since SLC1A5 ablation reduced intracellular amino acids, we hypothesized osteoblasts rely on glutamine and asparagine metabolism to maintain cellular amino acid pools. To test the validity of this hypothesis, we traced the relative contribution of glutamine or asparagine into amino acids directly. Consistent with previous reports, glutamine carbon contributes to all TCA cycle intermediates and was significantly enriched in several amino acids (e.g. glutamate, aspartate, alanine, and proline) found to be reduced in *Slc1a5* targeted cells (*Figure 6A*, *Figure 6—figure supplement 1A*). Similarly, glutamine nitrogen was significantly enriched in glutamate, aspartate, alanine, serine, glycine and proline (*Figure 6B*). Consistent with the minor effects of asparagine withdrawal on markers of amino acid depletion, asparagine carbon was enriched only in aspartate, malate, fumarate, and citrate (*Figure 6A*, *Figure 6—figure supplement 1A*). On the other hand, asparagine nitrogen was enriched in aspartate, glutamate, proline, serine, and alanine (*Figure 6A–B*). Thus, glutamine contributes both carbon and nitrogen for amino acid biosynthesis whereas asparagine carbon is used only for aspartate biosynthesis while asparagine nitrogen is used in transamination reactions. Importantly, the amino acids derived from either glutamine carbon (e.g. Glu, Ala, Asp, Pro) and nitrogen (e.g. Glu, Ala, Asp, Ser, Gly and Pro) and asparagine carbon (e.g. Asp) and nitrogen (e.g. Asp, Pro, Ala, Ser) were significantly enriched in total protein (*Figure 6C–D*). Rescue experiments found NEAA did not rescue proliferation but could rescue the induction of terminal osteoblast marker genes like *Ibsp* and to a lesser extent *Bglap* in *Slc1a5*-deficient cells (*Figure 4—figure supplement 1*). These data indicate *Slc1a5* provides glutamine and asparagine that are used for de novo synthesis of many amino acids (e.g. Glu, Asp, Ala, Ser, Gly, and Pro) that can be incorporated into nascent protein in osteoblasts.

Next, we sought to determine if glutamine-dependent amino acid synthesis was required for the high rate of protein synthesis in osteoblasts. We focused on glutamine because glutamine was more

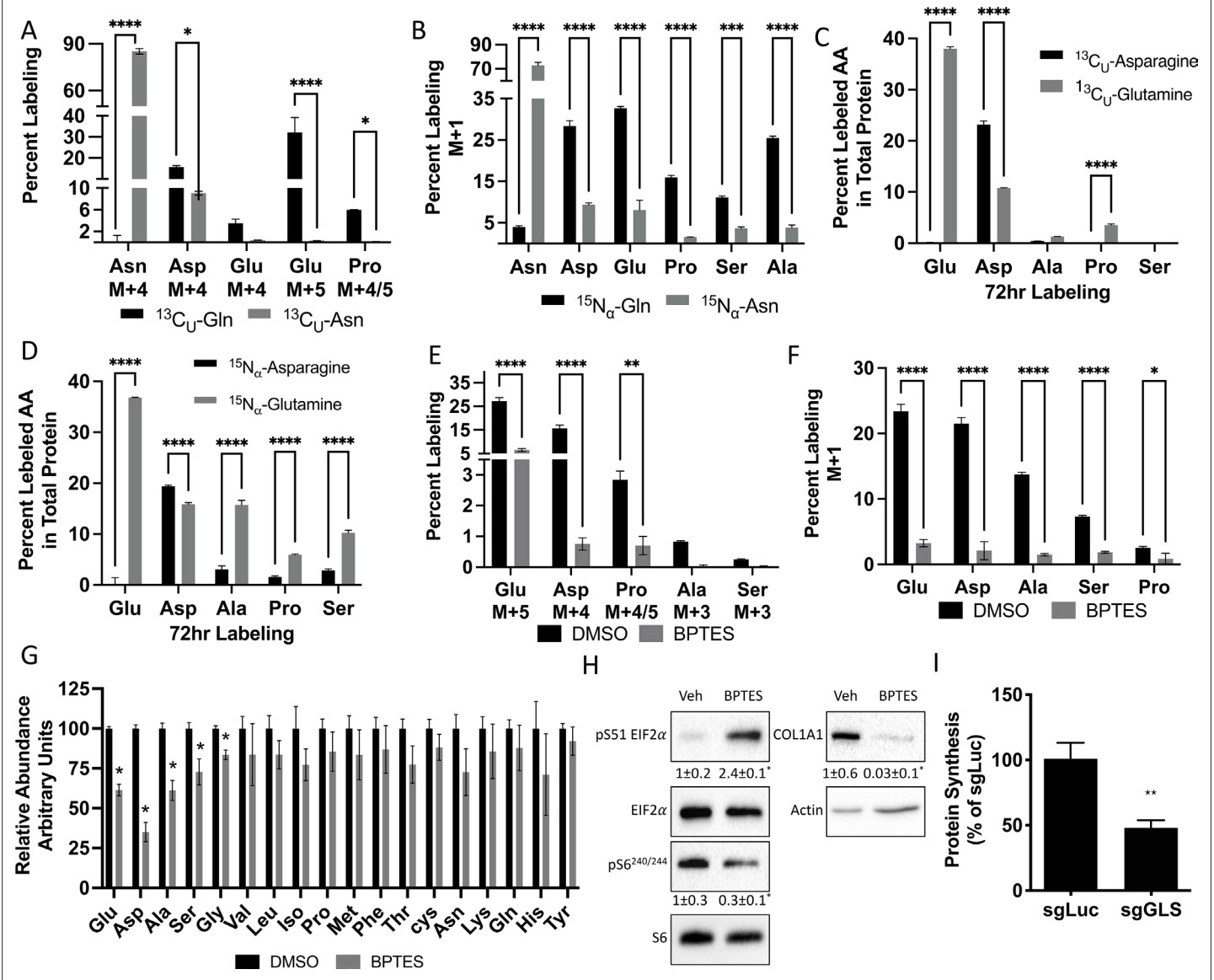

**Figure 6.** Glutamine and asparagine are utilized for de novo amino acid biosynthesis. (**A–B**) Fractional contribution of [U-$^{13}$C]glutamine or [U-$^{13}$C] asparagine (**A**) or [α–$^{15}$N]glutamine or [α–15N]asparagine (**B**) to asparagine, aspartate, glutamate, proline, serine, and alanine. (**C–D**) Fractional contribution of [U-$^{13}$C]glutamine or [U-$^{13}$C]asparagine (**C**) or [α–15N]glutamine or [α–15N]asparagine (**D**) to asparagine, aspartate, glutamate, proline, serine, and alanine in total protein. (**E–F**) Effect of BPTES treatment on the fractional contribution of [U-$^{13}$C]glutamine (**E**) or [α–15N]glutamine (**F**) to amino acids. * p ≤ 0.05, ** p ≤ 0.005, *** p ≤ 0.0005, **** p ≤ 0.00005 by ordinary one-way ANOVA with Tukey's multiple comparisons. (**G**) Effect of GLS inhibition on intracellular amino acid concentration measured by mass spectrometry. * p ≤ 0.05, multiple unpaired *t*-tests. Error bars depict SD.(**H**) Western blot analyses of the effects of BPTES treatment on mTORC1 signaling, Eif2a phosphorylation and COL1A1 expression. Phospho-proteins normalized to respective total protein. COL1A1 normalized to beta-actin. Fold change± SD for three independent experiments. * p ≤ 0.05 by an unpaired two-tailed Student's *t*-test. (**I**) Effect of GLS inhibition on protein synthesis as determined by the rate of $^{3}$H Proline incorporation into total protein. * p ≤ 0.05 by an unpaired two-tailed Student's *t*-test. Error bars depict SD.

The online version of this article includes the following figure supplement(s) for figure 6:

**Source data 1.** Contains numerical and uncropped western blot source data for *Figure 6*.

**Figure supplement 1** Inhibiting glutamine metabolism recapitulates molecular effects of SLC1A5 ablation.

**Figure supplement 1—source data 1.** Contains numerical and uncropped western blot source data for *Figure 6—figure supplement 1*.

widely used for amino acid synthesis, and glutamine withdrawal more completely phenocopied the cellular effects of *Slc1a5* targeting. To do this, we inhibited the enzyme glutaminase (GLS) which catalyzes the first rate-limiting step in glutamine metabolism. GLS inhibition using BPTES resulted in cellular effects similar to *Slc1a5* ablation. GLS inhibition significantly reduced glutamine carbon and nitrogen contribution to amino acid synthesis and reduced the intracellular concentrations of glutamate, aspartate, alanine, serine, and glycine concentrations (*Figure 6E–G*). Consistent with reduced amino acid concentrations, GLS inhibition increased the presence of uncharged Glu, Asp, and Pro tRNA without affecting either Gln or Val tRNA charging (*Figure 6—figure supplement 1D*). Finally, GLS inhibition induced eIF2α Ser51 phosphorylation and reduced S6 Ser240/244 phosphorylation (*Figure 6H*). Consistent with these molecular changes, GLS inhibition significantly reduced EdU incorporation and COL1A1 expression (*Figure 6H–I*). Importantly, *Gls* targeting reduced overall protein synthesis like *Slc1a5* targeting (*Figure 6I*, *Figure 6—figure supplement 1E-G*). These data highlight the importance of de novo amino acid synthesis to maintain amino acid homeostasis and promote proliferation and protein synthesis in osteoblasts.

## Discussion

Here, we present data demonstrating that SLC1A5 is a critical regulator of osteoblast proliferation, differentiation and bone formation. Osteoblasts increase *Slc1a5* expression as they undergo differentiation and increase bone matrix production. Genetically inhibiting SLC1A5 activity in preosteoblasts results in delayed endochondral and intramembranous ossification. Delayed bone development is the result of decreased osteoprogenitor proliferation as well as a reduction in overall osteoblast differentiation. *Slc1a5* deficient osteoblasts have reduced protein synthetic activity which manifests in reduced OSX and COL1A1 protein expression. COL1A1 is the major organic component of bone matrix and OSX is required for terminal osteoblast differentiation (*Nakashima et al., 2002*). Mechanistically, SLC1A5 primarily provides glutamine and to a lesser extent asparagine in osteoblasts.

Glutamine is an essential nutrient in osteoblasts. We and others have shown that glutamine metabolism contribute to many downstream metabolites in osteoblasts including αKG, non-essential amino acids (NEAA) and the tripeptide antioxidant glutathione (*Karner et al., 2015*; *Yu et al., 2019*; *Stegen et al., 2021*). These different studies have demonstrated these glutamine derived metabolites regulate distinct processes during osteoblast differentiation. For example, αKG is essential for skeletal stem cell proliferation, GSH regulates osteoblast viability while NEAA are important for differentiation (*Yu et al., 2019*; *Stegen et al., 2021*). We have extended these previous findings by identifying SLC1A5 as the primary glutamine transporter in osteoblasts. SLC1A5 is responsible for the majority of glutamine uptake importing 53.5 % of glutamine in osteoblasts. In addition, *Slc1a5* is a minor contributor to asparagine uptake importing only 19.5 % of asparagine in osteoblasts (*Figure 4B*). Consistent with our own and others previous reports, we found glutamine is a critical nutrient at all stages of osteoblast differentiation whereas asparagine is more important for terminal osteoblast differentiation. To our knowledge, this represents the first demonstration of the necessity of asparagine for osteoblast differentiation. Although, osteoblasts likely rely heavily on other transporters in addition to SLC1A5 to provide asparagine during differentiation.

Osteoblasts increase SLC1A5 expression to supply glutamine and asparagine which are used to maintain amino acid homeostasis. Interestingly, glutamine was more widely used for NEAA biosynthesis as our tracing experiments found glutamate, alanine, serine, aspartate, proline, and glycine were all synthesized from either glutamine carbon or nitrogen. Conversely, only aspartate was synthesized using asparagine carbon whereas Asp, Glu, Ala, and Ser were synthesized using asparagine nitrogen albeit at lower levels compared to glutamine (*Figure 6A–B*). It is important to note that while we focused our analyses on amino acid biosynthesis, our data does not preclude a role for glutamine and asparagine to support osteoblast proliferation or differentiation by providing other metabolites. Consistent with this, NEAA could not rescue proliferation in *Slc1a5* deficient cells (*Figure 4—figure supplement 1E*). Moreover, the functional significance of asparagine derived amino acids in osteoblasts remains enigmatic. The effects of asparagine depletion on osteoblast differentiation may be due to direct effects on protein synthesis, reduction of select asparagine sensitive proteins or loss of an as yet undefined asparagine derived metabolite (e.g. nucleotides). Future studies into the role and downstream metabolism of asparagine in osteoblasts are warranted.

It is interesting that glutamine and asparagine metabolism impinge upon aspartate biosynthesis in osteoblasts. Aspartate synthesis is critical for proliferation by providing both carbon and nitrogen for synthetic reactions (*Birsoy et al., 2015*; *Sullivan et al., 2015*). Thus, glutamine (and asparagine when glutamine is limiting) likely regulate proliferation in part by providing both carbon and nitrogen for aspartate biosynthesis (*Pavlova et al., 2018*). Aspartate synthesis likely occurs as part of the malate aspartate shuttle which is critical to regenerate oxidized $NAD^+$ to maintain high rates of glycolysis in osteoblasts (*Lee et al., 2020*). While we do not completely understand the metabolism of aspartate in osteoblasts, the aspartate derived from both glutamine and asparagine was enriched in total protein (*Figure 6E*). Interestingly, while glutamine contributed more to aspartate biosynthesis than asparagine, asparagine-derived aspartate was more highly enriched in total protein. This suggests disparate pools of aspartate exist in osteoblasts and that the metabolic fates for these glutamine- or asparagine-derived aspartate pools vary. For example, glutamine derived aspartate may contribute more to nucleotide biosynthesis to facilitate osteoblast proliferation whereas asparagine derived aspartate is used more for protein and matrix synthesis.

Proliferation is one of the initial stages of the osteoblast differentiation process. Proliferation is associated with increased demand for glucose and amino acids to support both nucleotide and NEAA biosynthesis required for cell division (*Guntur et al., 2014*; *Karner and Long, 2018*; *Flanagan and Nichols, 1964*; *Hosios et al., 2016*). To fulfil this need, proliferating cancer cells increase amino acid transporter expression to enhance amino acid supply (*Bhutia et al., 2015*; *Liu et al., 2018*; *Hassanein et al., 2013*; *DeBerardinis et al., 2008*; *Jones et al., 2018*; *Kiyokawa et al., 1996*; *Krall et al., 2016*). Here, we find that *Slc1a5* provides glutamine and asparagine to osteoblasts, but that only glutamine is required for proliferation. This is consistent with recent reports that asparagine is important for proliferation only when glutamine is limited (*Pavlova et al., 2018*; *Vander Heiden and DeBerardinis, 2017*; *Zhang et al., 2014*). Glutamine is an essential nutrient in proliferating cells (*Eagle, 1955*), and recent studies found glutamine and its metabolism are critical for proliferation in osteoblast progenitor cells (*Yu et al., 2019*; *Stegen et al., 2021*). The precise mechanism by which SLC1A5 regulates proliferation remains unclear as amino acid supplementation was unable to rescue proliferation. This may be because NEAA uptake is inherently limited in osteoblasts resulting in the need for amino acid biosynthesis. Alternatively, glutamine catabolism provides amino acids and other intermediate metabolites (e.g. α-KG) to support proliferation. Consistent with this, we and others have shown that either α-KG or NEAA can rescue proliferation in GLS-deficient skeletal stem cells or osteoblasts respectively (*Yu et al., 2019*; *Stegen et al., 2021*).

During differentiation, osteoblasts increase their capacity for protein synthesis and secretion. Here, we find that *Slc1a5* expression is required for robust protein synthesis in osteoblasts. This effect is likely regulated by multiple factors. First, *Slc1a5* provides glutamine and asparagine that are likely directly incorporated into nascent protein. Second, our tracing data shows both glutamine and asparagine are used for de novo amino acid synthesis to support protein synthesis. Third, *Slc1a5* provides glutamine to activate the mechanistic target of rapamycin complex 1 (mTORC1), an important regulator of both osteoblast proliferation and differentiation (*Nicklin et al., 2009*; *Singha et al., 2008*; *Xian et al., 2012*; *Chen et al., 2014*; *Fitter et al., 2017*). mTORC1 is tightly regulated by amino acid availability and functions to increase protein synthesis when amino acids are available. Consistent with this, *Slc1a5*-deficient osteoblasts had decreased mTORC1 activation and reduced protein synthesis (*Figures 3A–B , and 4C*). Finally, *Slc1a5*-deficient osteoblasts were characterized by robust activation of the integrated stress response (ISR). When amino acids are limiting, uncharged tRNA activates the kinase general control nonderepressible 2 (GCN2) which phosphorylates eukaryotic initiation factor 2 a (eIF2a) to attenuate global protein synthesis (*Zhang et al., 2002*; *Ye et al., 2010*; *Kroemer et al., 2010*; *Pakos-Zebrucka et al., 2016*). These molecular changes in *Slc1a5*-deficient osteoblasts are primarily the result of reduced glutamine uptake as our drop out experiments found glutamine but not asparagine withdrawal reduced mTORC1 activation and increased ISR like *Slc1a5* ablation (*Figure 5C*). Moreover, inhibiting glutamine catabolism similarly affected mTORC1 and ISR activation (*Figure 6H*). Thus, *Slc1a5* provides glutamine used to maintain amino acid homeostasis to regulate protein synthesis directly and indirectly downstream of mTORC1 and GCN2-dependent ISR.

In summary, we have defined the necessity and the molecular substrates of *Slc1a5* in osteoblasts. Our data indicates that SLC1A5 is the primary glutamine transporter in osteoblasts. In addition to glutamine, SLC1A5 acts cell autonomously in osteoblasts to provide asparagine, both of which are

essential for osteoblast differentiation. These data highlight an elegant mechanism by which osteoblasts both directly obtain and synthesize the requisite amino acids to support osteoblast differentiation and bone formation.

# Materials and methods

**Key resources table**

| Reagent type (species) or resource | Designation | Source or reference | Identifiers | Additional information |
|---|---|---|---|---|
| Genetic Reagent (*M. musculus*) | C57Bl/6 J | Jackson Laboratory | RRID:IMSR_JAX:000664 | |
| Genetic Reagent (*M. musculus*) | *Rosa26Cas9* | Jackson Laboratory | RRID:IMSR_JAX:024858 | |
| Genetic Reagent (*M. musculus*) | *Rosa26Flpe* | Jackson Laboratory | RRID:IMSR_JAX:003946 | |
| Genetic Reagent (*M. musculus*) | *Sp7tTA;tetOeGFP/Cre* | PMID:16854976 | RRID:IMSR_JAX:006361 | |
| Genetic Reagent (*M. musculus*) | *Slc1a5$^{flox}$* | This paper | | See **Methods – Mouse strains** for more information. |
| Chemical compound, drug | Ascorbic acid | Sigma | Cat#A4544 | |
| Chemical compound, drug | β-glycerophosphate | Sigma | Cat#G9422 | |
| Chemical compound, drug | one-step NBT/BCIP solution | Thermofisher | Cat#PI34042 | |
| Chemical compound, drug | L-(2,3,4-$^3$H)Glutamine | Perkin Elmer | Cat#NET551250UC | |
| Chemical compound, drug | L-($^{14}$C)Alanine | Perkin Elmer | Cat#EC266E250UC | |
| Chemical compound, drug | L-(2,3,-$^3$H)Proline | Perkin Elmer | Cat#NET483250UC | |
| Chemical compound, drug | L-(2,3,-$^3$H)Aspartic acid | Perkin Elmer | Cat#NET390V001MC | |
| Chemical compound, drug | L-(4,5-$^3$H(N))Lysine | Perkin Elmer | Cat#NEC280E050UC | |
| Chemical compound, drug | L-(2,3,$^3$ H)Asparagine | American Radiolabeled Chemicals | Cat#ART-0500–250 | |
| Chemical compound, drug | [U-$^{13}$C]glutamine, | Sigma | Cat#605,166 | Used at 2 mM final concentration |
| Chemical compound, drug | [α-$^{15}$N]glutamine | Sigma | Cat#486,809 | Used at 2 mM final concentration |
| Chemical compound, drug | [U-$^{13}$C]Asparagine | Sigma | Cat#579,866 | Used at 0.33 mM final concentration |
| Chemical compound, drug | [α-$^{15}$N] Asparagine | Sigma | Cat#485,896 | Used at 0.33 mM final concentration |
| Chemical compound, drug | AP substrate BM purple | Roche | Cat#11442074001 | |
| Chemical compound, drug | ECL substrate | Biorad | Cat#1705060 | |
| Chemical compound, drug | super signal West Femto ECL. | Thermofisher | Cat#1705060 | |
| Antibody | Eif2α (Rabbit monoclonal) | Cell Signaling | RRID:AB_10692650 | (1:1000) |
| Antibody | pSer51 Eif2α (Rabbit monoclonal) | Cell Signaling | RRID:AB_2096481 | (1:1000) |
| Antibody | pSer240/244 S6rp(rabbit polyclonal) | Cell Signaling | RRID:AB_331682 | (1:1000) |
| Antibody | S6rp (rabbit monoclonal) | Cell Signaling | RRID:AB_331355 | (1:1000) |
| Antibody | α-tubulin (rabbit monoclonal) | Cell Signaling | RRID:AB_2619646 | (1:1000) |
| Antibody | α-actin (rabbit polyclonal) | Cell Signaling | RRID:AB_330288 | (1:2000) |
| Antibody | HRP goat anti-rabbit (goat polyclonal) | Cell Signaling | RRID:AB_2099233 | (1:2000) |

*Continued on next page*

*Continued*

| Reagent type (species) or resource | Designation | Source or reference | Identifiers | Additional information |
|---|---|---|---|---|
| Antibody | HRP anti-mouse (horse polyclonal) | Cell Signaling | RRID:AB_330924 | (1:2000) |
| Antibody | COL1A1 (mouse monoclonal) | Santa Cruz | RRID:AB_2797597 | (1:2000) WB (1:1000) IF |
| Antibody | OSX (rabbit polyclonal) | Abcam | RRID:AB_2194492 | (1:1000) |
| Antibody | PCNA (mouse monoclonal) | Cell Signaling | RRID:AB_2160343 | (1:500) |
| Antibody | goat anti mouse 568 (goat unknown clonality) | Thermofisher | RRID:AB_141359 | (1:1000) |
| Antibody | goat anti Rabbit 568 (goat polyclonal) | Thermofisher | RRID:AB_143157 | (1:1000) |
| Commercial assay or kit | Iscript Reverse transcription kit | Biorad | Cat#1708841 | |
| Commercial assay or kit | SYBR green | Biorad | Cat#1725275 | |
| Commercial assay or kit | Click-iT EdU Cell Proliferation Imaging Kit | Invitrogen, | Cat#C10337 | |
| Commercial assay or kit | Click-iT EdU Alexa Fluor 488 Flow Cytometry Assay Kit | Invitrogen | Cat#C10420 | |
| Commercial assay or kit | Apoptosis Assay Kit (Cat# 22837). | AAT BIO | Cat# 22,837 | |
| Chemical compound, drug | AP substrate BM purple | Roche | Cat#11442074001 | |
| Software Algorithm | Graphpad 6 | https://www.graphpad.com/ | | |

## Mouse strains

C57Bl/6 J (RRID:IMSR_JAX:000664), *Rosa26Cas9* (RRID:1MSR_JAX:024858), *Sp7tTA;tetOeGFP/Cre* (RRID:IMSR_JAX:006361) (denoted in the text as *Sp7Cre*), and *Rosa26Flpe* (RRID:IMSR_JAX:003946) mouse strains were obtained from the Jackson Laboratory. The *Slc1a5$^{flox}$* mouse strain was generated by the Duke Cancer Institute Transgenic Core facility. Briefly, LoxP sites flanking exon two and a frt PGK neo-cassette were inserted in to the endogenous *Slc1a5* locus using homologous recombination (Fig. S1). The chimeric mice containing the targeting vector were then crossed to the *Rosa26Flpe* mouse to remove the neomycin cassette. All mice were housed at 23 °C on a 12 hr light/dark cycle and maintained on PicoLab Rodent Diet 290 (Lab diet#5053, St. Louis, MO). Timed pregnant females euthanized, and embryos were analyzed at E14.5, E15.5, E16.5, and P0. *Sp7tTA;tetOeGFP/Cre* mice expresses GFP and Cre recombinase under the control of the *Sp7* promoter. We evaluated GFP as a proxy for *Sp7* mRNA expression. The *Sp7tTA;tetOeGFP/Cre* mice have a partially penetrant bone phenotype. To control for this, in all genetic experiments, *Sp7Cre;Slc1a5$^{fl/fl}$* 'knockout' mice are always compared to *Sp7Cre;Slc1a5$^{fl/+}$* as 'wild type' littermate controls. All animal studies were approved by the animal studies committees at Duke University and the University of Texas Southwestern Medical Center at Dallas.

## Cell culture

P3 pups were euthanized and the parietal and frontal bones were isolated according to standard protocols. The membranous tissue was removed, and the bones were washed and cleaned with PBS. The calvaria was chopped and four sequential 10 min digestions were performed with 1 mg/ml Collagenase P in a shaking incubator at 37 °C. The first digestion was discarded, and the rest were combined. The digested calvarial cells were centrifuged and plated with α-MEM (GIBCO) supplemented with 15 % FBS (Invitrogen). The cells were seeded at 50,000 cells/ml. To initiate osteoblast differentiation the growth media was replaced with α-MEM with 10 % FBS, 50 mg/ml ascorbic acid (Sigma) and 10 mM β-glycerophosphate (Sigma). RNA was isolated at day 7 after mineralization media was added. At day 14, alizarin red and Von kossa staining was performed to visualize matrix mineralization. Alkaline phosphatase staining was performed using the one-step nitro-blue tetrazolium (NBT) and 5-bromo-4-chloro-3'-indolyphosphate p-toluidine salt (BCIP) solution (Thermofisher). For amino acid rescue experiments, cells were cultured for either 24 hr (proliferation) or 7 days in α-MEM supplemented

**Table 1.** sgRNA sequences.

| Slc1a5.g4 | ATTGATCTCCCGCTGGATACNGG |
|---|---|
| Slc1a5.g1 | ACCCGTTGGAATCCTGTTCCNGG |
| Slc1a5.g10 | AAAATCCCTATCGATTCCTGNGG |
| Slc1a5.g17 | AGAAGAGGTCCCGAAAGCAGNGG |
| Slc1a5.g31 | CCAGGAGCCCGTGGATGGCGNGG |
| MS344.Gls.g5 | ATATAACTCATCGATGTGTGNGG |
| MS344.Gls.g3 | GTGCTAAAAAGCAGTCTGGANGG |
| MS345.Gls.g3 | CAAATTCAGTCCTGATTTGTNGG |
| MS346.Gls.g14 | ATATTTCAGGGGTTTTACACNGG |
| MS346.Gls.g1 | TGCAATTGCTGTTAATGACCNGG |
| SP498.mCherry.g17 | CAAGTAGTCGGGGATGTCGGNGG |
| SP498.mCherry.g19 | AGTAGTCGGGGATGTCGGCGNGG |
| SP499.Luc.g3 | CAATTCTTTATGCCGGTGTTNGG |
| SP399.Luc.g4 | GTGTTGGGCGCGTTATTTATNGG |

to the indicated concentration with the following amino acids: 0.67 mM glycine, 0.28 mM alanine, 0.23 mM aspartic acid, 0.2 mM histidine, 0.24 mM serine, 0.35 mM proline, 0.4 mM lysine, and 0.5 mM arginine.

CRISPR/Cas9 targeting sgRNA vectors were purchased from the Genome Engineering and iPSC Center at Washington University School of Medicine. cOB were isolated from *Rosa*[Cas9] mice and were infected with five lentiviral delivered short guide RNAs targeting exons 4–6 of either *Slc1a5* (sgSlc1a5) or *Gls* (sgGls) (Fig. S1 and S6). As a control cOB were infected with sgRNAs targeting the open-reading frames of Luciferase and mCherry as a control (designated sgLuc in text and Fig.s). sgRNA sequences are listed in *Table 1*. To make virus, 293T cells were cotransfected with the lentiviral vector expressing short guide RNAs, pMD2.g and psPax2. After 48 hr of transfection, the media containing the virus was collected and filtered through 0.45µm filter. cOB were cultured to 50 % confluency and were infected for 24 hr followed by recovery in regular media for another 24 hr.

## Uptake assays and proline incorporation assay

Confluent primary cells were washed ones with PBS and two additionally washes with Krebs Ringer Hepes (KRH) (120 mM NaCl, 5 mM KCl, 2 mM CaCl$_2$, 1 mM MgCl$_2$, 25 mM NaHCO$_3$, 5 mM HEPES, 1 mM D-Glucose, pH 8). Cells were then treated for 5 min with KRH containing 4 µCi/mL of either L-(2,3,4-$^3$H)Glutamine, L-($^{14}$C)Alanine, L-(2,3,-$^3$H)Proline, L-(2,3,-$^3$H)Aspartic acid, L-(4,5-$^3$H(N))Lysine, or L-(2,3,-$^3$H)Asparagine. After 5 min, the uptake was terminated by washing the cells three times with ice cold KRH. Cells were then scraped in 1 mL of dH$_2$O and lysed by sonication for 1 min with 1 second pulses at 20 % amplitude. The lysate was then centrifuged and mixed with 8 mL scintillation liquid. The counts per minute (CPM) were measured using a Beckman LS6500 scintillation counter. Proline incorporation assay was performed on confluent cells in a 12-well plate. The cells were washed ones with PBS and two additional washes with KRH. The cells were incubated with 4 µCi/mL L (2,3,-$^3$H) Proline in KRH at 37 C for 3 hr. The uptake was terminated with three washes of cold KRH, and the cells were scraped with 150 µl RIPA. The lysates were then cleared and 90 µl of the protein was precipitated with 10 % tricholoroacetic acid (TCA). The pellet was washed three times with 5%TCA and resuspended with 1 ml 0.2 N NaOH. A total of 100 µl of the lysate was saved to measure proline incorporation in total protein. The remaining lysate was separated in to two 400 µl parts, to one-part 200 µl of 60 µm HEPES containing 10 units of collagenase was added and to the second part only HEPES was added. The lysates were incubated at 37 C for 2 hr, followed by precipitation of undigested protein with 10%TCA. The supernatant was added to 8 ml of scintillation liquid to measure proline incorporation in collagen using a Beckman LS6500 scintillation counter.

**Table 2.** RT-PCR primer sequences.

| Gene symbol | Forward | Reverse |
| --- | --- | --- |
| Slc1a5 | TGGAGATGAAAGACGTCCGC | CAGGCAGGCTGACACTGGAT |
| β-actin | AGATGTGGATCAGCAAGCAG | GCGCAAGTTAGGTTTTGTCA |
| Akp2 | CCAACTCTTTTGTGCCAGAGA | GGCTACATTGGTGTTGAGCTTTT |
| Ibsp | CAGAGGAGGCAAGCGTCACT | GCTGTCTGGGTGCCAACACT |
| Sp7 | CCTTCTCAAGCACCAATGG | AAGGGTGGGTAGTCATTTGCATA |
| Runx2 | CCAACCGAGTCATTTAAGGCT | GCTCACGTCGCTCATCTTG |
| Bglap | CAGCGGCCCTGAGTCTGA | GCCGGAGTCTGTTCACTACCTTA |
| Asns | CAAGGAGCCCAAGTTCAGTAT | GGCTGTCCTCCAGCCAAT |

## RT-PCR

RNA was isolated using the Qiagen RNAeasy kit. 500 ng of RNA was reverse transcribed to cDNA using Iscript Reverse transcription kit (Biorad). The cDNA was diluted 1:10. qPCR reaction was setup using SYBR green (Biorad), 2µl of diluted cDNA and 0.1 µM primers in technical and biological triplicates. ABI Quantstudio three was used to run the qPCR. The PCR cycle were 95 °C for 3 min followed by 35 cycles of 95 °C for 10 s and 60 °C for 30 s. Gene expression was normalized to *ActB* followed by calculating relative expression using the $2^{-(\Delta\Delta Ct)}$ method. The list of primers is found in *Table 2*.

## Western blotting

To isolate protein, cells were scraped, or bones were pulverized in RIPA lysis buffer (50 mM Tris (pH 7.4), 15 mM NaCl, 0.5% NP-40) containing protease and phosphatase inhibitors (Roche). Protein concentration was measured using the BCA method and 10 µg of the protein was run on 10 % polyacrylamide gel. The protein was then transferred to a nitrocellulose membrane which was blocked for 1 hour at room temperature with 5 % milk in TBST (TBS, 0.1 % Tween20). The blots were then incubated with primary antibodies raised against ASCT2/SLC1A5 (RRID:AB_10621427, 1:1000), Eif2α (RRID:AB_10692650), pSer51 Eif2α (RRID:AB_2096481) pSer240/244 S6rp (RRID:AB_331682, 1:1000), S6rp (RRID:AB_331355, 1:1000), α-tubulin (RRID: AB_2619646, 1:1000), or β-actin (RRID: AB_330288, 1:2000) overnight at 4 C. On day 2, the membranes were washed three times 5 min each with TBST and incubated with appropriate secondary antibody HRP goat anti-rabbit (RRID:AB_2099233) or HRP anti-mouse (RRID:AB_330924) in 5 % milk in TBST for 1 hr at room temperature. The membranes were again washed three times with TBST and developed using clarity ECL substrate (Biorad) or super signal West Femto ECL.

## EdU incorporation and annexin V assay

5000 cells/well were seeded in a 96 well plate for 12–16 hr after which the cells were incubated with 10 µM EdU (5-ethynyl-2'-deoxyuridine) for 6 hr. EdU incorporation was performed using the instructions provided in the Click-iT EdU Cell Proliferation Imaging Kit (Invitrogen, C10337). EdU incorporation was also analyzed using Click-iT EdU Alexa Fluor 488 Flow Cytometry Assay Kit (Invitrogen, C10420) after 24 hr incubation with 10 µM EdU. The cells were then trypsinized, permeabilized and stained as per kit instructions. Cell viability assay was performed using the Apoptosis Assay Kit (Cat# 22837).

## Skeletal prep

Timed pregnant females were euthanized, and pups were harvested at E14.5, E15.5, and P0. The embryos were skinned, eviscerated and dehydrated in 95 % ethanol overnight. The mice were transferred to acetone for another night to dissolve fat tissue. After which, the tissue was stained with Alcian blue 8 GX (0.03%, m/v in 70 % ethanol) and Alizarin red S (0.005%, m/v in dH$_2$O) solution containing 10 % acetic acid and 80 % ethanol. The stained skeletons were cleared in 1 % KOH followed by a gradient of glycerol.

## Histology, immunofluorescence, and in situ hybridization

Limbs and skulls from E15.5, E16.5, and P0 were harvested skinned and fixed overnight in 4 % PFA (paraformaldehyde). Limbs from E16.5 and P0 were decalcified in 14 % EDTA overnight, followed by transferring them to 30 % sucrose for frozen embedding in OCT and 70 % ethanol for paraffin embedding. Paraffin embedded blocks were sectioned at 5 μM thickness and utilized for Alcian blue and picro-sirius, von Kossa and alcian blue using standard protocols.

Immunofluorescence was performed on 10 μM frozen sections brought to room temperature and incubated with 3 % $H_2O_2$ for 10 min. The sections were then incubated with 1.5 % goat serum in PBST (PBS with 0.1 % Tween20) at room temperature for 1 hr. The sections were then incubated overnight in primary antibodies COL1A1 (RRID:AB_638601,1:200), OSX (RRID:AB_2194492 1:500,), PCNA (RRID:AB_2160343,1:500) diluted in 1.5 % goat serum in PBST. On day 2, the sections were washed three times with PBST for 5 min each and incubated with goat anti-mouse 568 (RRID:AB_141359, 1:1000) for COL1A1 and goat anti-Rabbit 568 (RRID:AB_143157, 1:1000) for OSX and PCNA for 30 min at room temperature. The sections were then mounted with DAPI and imaged. COL1A1 intensity was measured in the $Sp7^{GFP}$-positive region. OSX, $Sp7^{GFP}$ copositive cells were counted using Image J.

In situ hybridization was performed on 10 μM frozen sections. The sections were fixed with 4 % PFA for 10 min at room temperature followed by 10 min in acetylation solution (1.3 % triethanolamine, 0.175 % HCl, 0.375 % acetic anhydride in $dH_2O$). The sections were then washed and incubated with hybridization buffer (50 % formamide (deionized), 5 X SSC, pH 4.5 (use citric acid to pH), 1 % SDS, 50 μg/mL yeast tRNA, 50 μg/mL heparin) for 2 hr in a humidified chamber. The excess hybridization buffer was removed and prewarmed probe diluted 1:10 was added to the slides, covered with parafilm and incubated at 60 C overnight. The slides were immersed in 5 X SSC to remove the parafilm and washed twice with 0.2 X SSC for 30 min at 60 C. After an additional wash at room temperature with 0.2 X SSC the slides were transferred to NTT (0.15 M NaCl, 0.1 M tris-Cl pH 7.5, 0.1 % tween 20) for 10 min at room temperature. The slides were blocked with blocking buffer (5 % heat inactivated sheep serum, 2 % blocking reagent/NTT) for 2 hr, followed by incubation with anti-Dig AP antibody diluted at 1:4000 in the blocking buffer overnight at 4 C. On the third day, the slides were washed with NTT on rotator three times for 30 min each, followed by 3 5 min washes with NTTML (0.15 M NaCl, 0.1 M tris pH 9.5, 50 mM $MgCl_2$,2mM Levamisole, 0.1 % tween 20). The slides were then incubated with prewarmed AP substrate BM purple (ROCHE) at 37 C and monitored for desired staining. After staining was achieved the slides were rinsed in PBS 3 times for 5 min each. The slides were fixed in 0.2 % glutaraldehyde in 4 % PFA overnight, the slides were then mounted with glycergel and imaged.

## Micro computed tomography (uCT)

Micro computed tomography (μCT) (VivaCT80, Scanco Medical AG) was used for three-dimensional reconstruction and analysis of bone parameters. Calvariae were harvested from either newborn mice or 2-month-old mice. All muscle and extemporaneous tissue were removed and the isolated calvariae were washed in PBS, fixed overnight in 10%NBF and dehydrated in 70 % ethanol. The calvariae were immobilized in 2 % agarose in PBS for scanning. A fixed volume surrounding the skull was used for 3D reconstructions. In newborn calvariae, bone volume was quantified from a fixed number of slices in the occipital lobe. The threshold was set at 280. For quantification of bone mass in the long bone, 2-month-old femurs were isolated, fixed, immobilized, and scanned. Bone parameters were quantified from 200 slices directly underneath the growth plate with the threshold set at 320.

Mass spectrometry and metabolic tracing cOB isolated from RosaCas9 homozygous P3 pups were cultured in 6 cm plates and treated as indicated. Cells were incubated with 2 mM [U-$^{13}$C]glutamine, 2 mM [α–15N]glutamine, 0.33 mM [U-$^{13}$C]Asparagine and 0.33 mM [α–15N] Asparagine for either 24 or 72 hours as indicated. Cells treated for 24 hr were washed with cold PBS and isolated three times with –80 °C methanol on dry ice in Eppendorf tubes containing 20 nM norvaline, which is an internal standard. The extracts were vortexed and centrifuged at 12,000 RPM for 15 min. The supernatant was completely dried with $N_2$ gas. The dried residue was resuspended in 25 μl methoxylamine hydrochloride (2 % MOX in pyridine) and incubated for 90 min at 40 °C on a heat block. After the incubation, the samples were spun at maximum speed for 2 min and 35 μl of MTBSTFA + 1 % TBDMS was added followed by a 30 min incubation at 60 °C. The samples were centrifuged at 12,000 RPM for 5 minutes and the supernatant was transferred to GC vials for GC-MS analysis. For the experiments

tracing labeled amino acids into protein, cells were incubated with 2 mM [U-$^{13}$C]glutamine, 2 mM [α–15N]glutamine, 0.33 mM [U-$^{13}$C]Asparagine or 0.33 mM [α–15N] Asparagine for 72 hours. Cells were isolated in 1 ml 1 M PCA and centrifuged for 10 min to precipitate the protein. The precipitate was washed twice with 1 ml 70 % ethanol after which 20 nM norvaline was added to the samples. The protein was hydrolyzed with 1 ml 6 M HCl at 110 °C for 18 hr. The samples were cooled and 1 ml chloroform was added and vortexed to remove hydrophobic metabolites. The isolates were centrifuged for 10 minutes and 100 μl of the supernatant was dried with $N_2$ gas until dry, 50 μl of MTBSTFA + 1 % TBDMS was added followed by a 30 min incubation at 60 °C. The samples were transferred to GC vials for GC-MS analysis.

### tRNA aminoacylation assay

The method is adapted from two recent reports (*Loayza-Puch et al., 2016*). Purified RNA was resuspended in 30 mM NaOAc/HOAc (pH 4.5). RNA was divided into two parts (2μ g each): one was oxidized with 50 mM NaIO$_4$ in 100 mM NaOAc/HOAc (pH 4.5) and the other was treated with 50 mM NaCl in NaOAc/HOAc (pH 4.5) for 15 min at room temperature. Samples were quenched with 100 mM glucose for 5 min at room temperature, followed by desaltation using G50 columns and precipitation using ethanol. tRNA was then deacylated in 50 mM Tris-HCl (pH 9) for 30 min at 37°C, followed by another ethanol precipitation. RNA (400 ng) was then ligated the 3'adaptor.

(5'-/5rApp/TGGAATTCTCGGGTGCCAAGG/3ddC/–3') using T4 RNA ligase 2(NEB) for 4 hr at 37 °C. 1μ g RNA was then reverse transcribed using SuperScript III first strand synthesis system with the primer (GCCTTGGCACCCGAGAATTCCA) following the manufacturer's instruction.

### Quantification and statistical analysis

All statistics were performed in Graphpad six software. In cell culture studies, statistical significance was determined by an unpaired two-tailed Student's $t$-test or one-way Anova. For uCT statistical significance was determine by a paired two-tailed Student's $t$-test comparing paired littermate controls. All quantifications are represented as mean ± standard deviation. A p value of less than 0.05 is considered statistically significant. All experiments were performed with n ≥ 3 biological replicates. The sample size and statistical analysis are noted in the Figure legends.

## Acknowledgements

The authors thank Drs. Vishal Patel and Guoli Hu for critical comments on this manuscript. This work was supported by National Institute of Health R01 grants (AR076325 and AR071967) to CMK.

## Additional information

### Funding

| Funder | Grant reference number | Author |
|---|---|---|
| National Institute of Arthritis and Musculoskeletal and Skin Diseases | AR076325 | Courtney M Karner |
| National Institute of Arthritis and Musculoskeletal and Skin Diseases | AR071967 | Courtney M Karner |

The funders had no role in study design, data collection and interpretation, or the decision to submit the work for publication.

### Author contributions

Deepika Sharma, Yilin Yu, Leyao Shen, Guo-Fang Zhang, Investigation, Writing – review and editing; Courtney M Karner, Conceptualization, Funding acquisition, Investigation, Project administration, Writing – original draft, Writing – review and editing

## Author ORCIDs
Courtney M Karner (iD) http://orcid.org/0000-0003-0387-4486

## Ethics
This study was performed in accordance with the recommendations in the Guide for the Care and Use of Laboratory Animals of the National Institutes of Health. All animal procedures were approved (APN 2020-102999) by the Animal Studies Committees at Duke University and the University of Texas Southwestern Medical Center at Dallas.

## Decision letter and Author response
Decision letter https://doi.org/10.7554/eLife.71595.sa1
Author response https://doi.org/10.7554/eLife.71595.sa2

## Additional files

### Supplementary files
• Transparent reporting form

### Data availability
All data generated or analyzed during this study are included in the manuscript.

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
