## [Decision Letter]

**Acceptance summary:**

This is an important work that evaluates the role of ASCT2 transporter in amino acid metabolisms in osteoblasts. State of the art in vitro and in vivo methods are used to prove that ASCT2 plays an important role in bone cells and tissue to supply glutamine required for normal osteoblasts function and matrix biosynthesis. This work adds new knowledge of osteoblast metabolism.

**Decision letter after peer review:**

Thank you for submitting your article "SLC1A5 provides glutamine and asparagine necessary for bone development in mice" for consideration by *eLife*. Your article has been reviewed by 3 peer reviewers, one of whom is a member of our Board of Reviewing Editors, and the evaluation has been overseen by a Reviewing Editor and Carlos Isales as the Senior Editor. The reviewers have opted to remain anonymous.

Essential revisions:

Reviewers all agree the study is interesting, but a series of critical issues were raised that need to be addressed by either performing additional experiments or clarifying their interpretation of the current results.

Please address concerns and comments point-by-point.

In particular, it is crucial to carefully discuss the novelty of the findings. In addition, it is critical to clarify whether the OSX-Cre driver used in the study is inducible.

*Reviewer #1 (Recommendations for the authors):*

Specific Comments:

1. The authors show that impairing the intracellular transport of glutamine and asparagine decreases the intracellular accumulation of metabolites of the TCA cycle such as citrate and α-ketoglutarate. Is mitochondrial respiration negatively affected in mutant osteoblasts? Are the mutant cells energy deficient?

2. Are the widening of the sutures and the delay of the osteogenic front still present in the flat bones of the skull in mutant mice postnatally?

*Reviewer #2 (Recommendations for the authors):*

1. The authors propose that inactivation of SLC1A5 affects osteoblast differentiation. While matrix mineralization is clearly decreased, it is not fully clear whether ALP staining is affected to the same extent using the different in vitro models (i.e. sgRNA against Slc1a5, cells from SLC1A5 conditional knockout mice, GPNA-treatment).

2. Based on the whole-skeleton stainings, the defect caused by SLC1A5 deletion appears to disappear in long bones of P0 mice. Is BV/TV decreased in long bones of SLC1A5 knockout mice, and, if so, does the bone phenotype progress in (young) adult mice? This is likely important information, as this would suggest that only a transient upregulation of SLC1A5 activity is required during bone development.

3. Is protein and matrix synthesis affected by GPNA treatment, thereby mimicking the genetic models?

4. The observation that asparagine depletion increases proliferation is interesting. Is glutamine uptake and/or catabolism increased in these conditions?

5. Figure 6A-F, Figure S5A-B – please provide statistics.

6. Please provide scale bars in Figure 1P-U, Figure S1Q-W, Figure S2A-W.

7. There appears to be some variation in the degree of mineralization in the different experiments (e.g. Figure S2Y), which might affect the effect size of the treatment condition.

8. For some experiments, the authors used the term 'ACTS2' instead of SLC1A5. It might be clearer to use SLC1A5 throughout.

*Reviewer #3 (Recommendations for the authors):*

This work by Sharma et al., studied the role of aa transporter, ASCT2, encoded by Slc1a5 gene, that transports mostly Glmn and Asn, in osteoblasts (OB). They use gene targeting in vitro and in vivo using Sp7-Cre driven cKO. They found that ASCT2 deletion impairs OB differentiation in vitro as well as mostly intramembranous ossification in vivo by interfering with proliferation and protein synthesis. Mechanistically, they show that Glmn uptake via ASCT2 is important for aa synthesis in OBs.

This group has shown before that Glmn is essential for OB metabolism. The current work further investigates this phenomenon and identifies ASCT2 as the key mechanism of Glmn uptake into OBs.

The work is logically structured and carefully done with appropriate in vivo and in vitro controls. A variety of methods is used to confirm their findings, such as in vivo immunodetection and in situ hybridization and in vitro metabolic tracing. The conclusions are well justified by the data.

---

## [Author Response]

Essential revisions:Reviewers all agree the study is interesting, but a series of critical issues were raised that need to be addressed by either performing additional experiments or clarifying their interpretation of the current results.Please address concerns and comments point-by-point.In particular, it is crucial to carefully discuss the novelty of the findings. In addition, it is critical to clarify whether the OSX-Cre driver used in the study is inducible.

We thank the editor and reviewers for their constructive comments that have given us the opportunity to improve this work. We can confirm the Sp7Cre driver (*Sp7Cre;tTAtetOeGFP*) we used in this study is inducible (DOX OFF) although we did not use it in this manner. We used this as a constitutive CRE line in this study. We have better described this Cre line and the controls we used in the methods. Also, we have expanded the discussion to include a new paragraph that better discusses the novelty of our findings. The relevant text is as follows:

“Glutamine is an essential nutrient in osteoblasts. We and others have shown that glutamine metabolism contribute to many downstream metabolites in osteoblasts including aKG, non-essential amino acids (NEAA) and the tripeptide antioxidant glutathione. These different studies have demonstrated these glutamine derived metabolites regulate distinct processes during osteoblast differentiation. For example, aKG is essential for skeletal stem cell proliferation, GSH regulates osteoblast viability while NEAA are important for differentiation. We have extended these previous findings by identifying SLC1A5 as the primary glutamine transporter in osteoblasts. SLC1A5 is responsible for a majority of glutamine uptake importing 53.5% of glutamine in osteoblasts. In addition, *Slc1a5* is a minor contributor to asparagine uptake importing 19.5% of asparagine in osteoblasts. Consistent with our own and others previous reports, we found glutamine is a critical nutrient at all stages of osteoblast differentiation whereas asparagine is more important for terminal osteoblast differentiation. To our knowledge this represents the first demonstration of the necessity of asparagine for osteoblast differentiation. Although it is likely that osteoblasts rely heavily on other transporters in addition to SLC1A5 to provide asparagine during differentiation.”

Reviewer #1 (Recommendations for the authors):Specific Comments:1. The authors show that impairing the intracellular transport of glutamine and asparagine decreases the intracellular accumulation of metabolites of the TCA cycle such as citrate and α-ketoglutarate. Is mitochondrial respiration negatively affected in mutant osteoblasts? Are the mutant cells energy deficient?

We thank the reviewer for the overall positive review of our work. We have not evaluated mitochondrial respiration directly. In our initial characterization of these mice, we did not observe obvious changes in markers of energetic stress (e.g. increased pThr172 of AMPK). Because of this, we focused our analyses on amino acid homeostasis as this was obviously affected in the *Slc1a5* mutant cells.

2. Are the widening of the sutures and the delay of the osteogenic front still present in the flat bones of the skull in mutant mice postnatally?

The suture phenotype is present in postnatal stages. We have included X-Ray, µCT and histology of 2-month-old *Sp7Cre;Slc1a5^fl/fl^* mice. Increased suture width and altered suture morphology is evident at this age. We have included these data in Figure 1 —figure supplement 2.

Reviewer #2 (Recommendations for the authors):1. The authors propose that inactivation of SLC1A5 affects osteoblast differentiation. While matrix mineralization is clearly decreased, it is not fully clear whether ALP staining is affected to the same extent using the different in vitro models (i.e. sgRNA against Slc1a5, cells from SLC1A5 conditional knockout mice, GPNA-treatment).

It is difficult to directly compare these experiments. One involves a chemical inhibitor (GPNA) while the other two are genetic in nature. Sp7Cre (which deletes Slc1a5 only in Sp7 expressing cells that already express ALPL) is not expressed in naïve calvarial cells and only becomes expressed after osteogenic induction. The sgRNA experiment is targeting Slc1a5 in naïve cells prior to ALPL and Sp7 expression. Likewise, GPNA is an acute inhibition targeting naïve cells. All three models demonstrated a consistent inhibition of both matrix mineralization and the induction of terminal osteoblast marker genes like *Ibsp* and *Bglap.* Thus, we conclude Slc1a5 is essential for osteoblast differentiation.

2. Based on the whole-skeleton stainings, the defect caused by SLC1A5 deletion appears to disappear in long bones of P0 mice. Is BV/TV decreased in long bones of SLC1A5 knockout mice, and, if so, does the bone phenotype progress in (young) adult mice? This is likely important information, as this would suggest that only a transient upregulation of SLC1A5 activity is required during bone development.

We apologize for giving the impression that the phenotype only affected the calvarium and the long bones were normal at birth. We evaluated the long bone phenotypes in the Slc1a5 knockout mice at both birth (P0) and 2-months of age. At birth, these mice are characterized by reduced expression of late osteoblast genes *Spp1* and *Bglap* with no change in *Col1a1* as measured by qPCR from whole bone shafts and in situ hybridization. At 2-months of age, the knockout mice have significantly decreased trabecular BV/TV and decreased OCN positive cells. We have included these data in Figure 2 —figure supplement 2.

3. Is protein and matrix synthesis affected by GPNA treatment, thereby mimicking the genetic models?

Yes, protein synthesis is significantly reduced in GPNA treated cells as determined by ^35^S-Cys and Met incorporation. This data is included in Figure 2 —figure supplement 1Y.

4. The observation that asparagine depletion increases proliferation is interesting. Is glutamine uptake and/or catabolism increased in these conditions?

The increase in proliferation when Asn is limiting was unexpected and we agree this finding is interesting. We have not evaluated either glutamine uptake or catabolism under these conditions. These are interesting experiments to perform in the future.

5. Figure 6A-F, Figure S5A-B – please provide statistics.

We have added statistics for these experiments.

6. Please provide scale bars in Figure 1P-U, Figure S1Q-W, Figure S2A-W.

We have added scale bars to these figures.

7. There appears to be some variation in the degree of mineralization in the different experiments (e.g. Figure S2Y), which might affect the effect size of the treatment condition.

We agree with the reviewer that the data in the original Figure S2Y is suboptimal. GPNA is dissolved in HCl, thus our vehicle control for those experiments is equal molar HCl. In our experience, this limits matrix mineralization under “control” conditions. Because the genetic data in Figures 1 and 2 is more robust we have removed the GPNA functional differentiation data.

8. For some experiments, the authors used the term 'ACTS2' instead of SLC1A5. It might be clearer to use SLC1A5 throughout.

Thank you for this suggestion. We have removed the references to ASCT2 and replaced them with SLC1A5.